# Leveraging paired serology to estimate the incidence of typhoidal *Salmonella* infection in the STRATAA study

Jo Walker[1]*, Paula Russell[2,3], Leanne Kermack[2,3], Tan Trinh Van[4], Tran Vu Thieu Nga[4], Elli Mylona[2,3], Susana Camara[5], Young Chan Kim[6], Sonu Shrestha[6], Arne Gehlhaar[6], Josefin Bartholdson Scott[2,3], Farhana Khanam[7], Mila Shakya[8,9], Deus Thindwa[10,11], Melita A. Gordon[10,11,12], Buddha Basnyat[7,13], John D. Clemens[6,14,15,16], Firdausi Qadri[6], Robert S. Heyderman[9,17], Christiane Dolecek[18], Susan Tonks[5], Thomas C. Darton[19], Andrew J. Pollard[5,18], Stephen Baker[2,3], James E. Meiring[9,10,11☯], Merryn Voysey[5,18☯], Virginia E. Pitzer[1☯], and the STRATAA Study Group[¶]

1 Department of Epidemiology of Microbial Diseases, Yale School of Public Health, New Haven, Connecticut, United States of America, 2 Cambridge Institute of Therapeutic Immunology and Infectious Disease, University of Cambridge School of Clinical Medicine, Cambridge, United Kingdom, 3 Department of Medicine, University of Cambridge School of Clinical Medicine, Cambridge, United Kingdom, 4 The Hospital for Tropical Diseases, Wellcome Trust Major Overseas Programme, Oxford University Clinical Research Unit, Ho Chi Minh City, Vietnam, 5 Oxford Vaccine Group, Department of Paediatrics, University of Oxford, Oxford, United Kingdom, 6 International Centre for Diarrhoeal Disease Research, Bangladesh, Dhaka, Bangladesh, 7 Oxford University Clinical Research Unit, Patan Academy of Health Sciences, Kathmandu, Nepal, 8 Patan Academy of Health Sciences, Patan Hospital, Lalitpur, Nepal, 9 Malawi-Liverpool-Wellcome Research Programme, Blantyre, Malawi, 10 College of Medicine, University of Malawi, Blantyre, Malawi, 11 Institute of Infection, Veterinary and Ecological Sciences, University of Liverpool, Liverpool, United Kingdom, 12 Nuffield Department of Medicine, Centre for Tropical Medicine and Global Health, University of Oxford, Oxford, United Kingdom, 13 UCLA Fielding School of Public Health, University of California Los Angeles, California, United States of America, 14 International Vaccine Institute, Seoul, South Korea, 15 Korea University Vaccine Innovation Center, Seoul, South Korea, 16 Research Department of Infection, Division of Infectious Diseases, University College London, London, United Kingdom, 17 Centre for Tropical Medicine and Global Health, Nuffield Department of Medicine, University of Oxford, Oxford, United Kingdom, 18 NIHR Oxford Biomedical Research Centre, Oxford, United Kingdom, 19 Clinical Infection Research Group, School of Medicine & Population Health, University of Sheffield, Sheffield, United Kingdom

☯ These authors contributed equally to this work.
¶ Membership of the STRATAA Study Group is provided in the Acknowledgements
* jo.walker@yale.edu

## Abstract

Serologic surveillance of at-risk populations can be used to directly estimate the incidence of typhoidal *Salmonella* infection across a variety of settings, including those without access to facility-based blood-culture surveillance. We collected paired blood samples approximately three months apart from an age-stratified random sample of healthy children and adults in Bangladesh, Malawi, and Nepal as part of the Strategic Typhoid Alliance Across Asia and Africa (STRATAA) study. We used a multiplex bead assay to measure the concentration of IgG antibodies against seven *Salmonella* typhi/paratyphi antigens (CdtB, FliC, HlyE, LPSO2, LPSO9, Vi, and YncE) in each sample and identified recently infected participants by fitting a regression

**Data availability statement:** Code and data are available at github.com/pitzerlab/Typhoid-Seroincidence.

**Funding:** The STRATAA study was funded by a Wellcome Trust Strategic Award (106158/Z/14/Z [PR, LK, TTV, TVTN, EM, SC, YCK, SS, AG, JBS, FK, MS, MAG, BB, JDC, FQ, RSH, CD, ST, TCD, AJP, SB, JEM, MV, VEP]) and the Bill & Melinda Gates Foundation (OPP1141321 [PR, LK, TTV, TVTN, EM, SC, YCK, SS, AG, JBS, FK, MS, MAG, BB, JDC, FQ, RSH, CD, ST, TCD, AJP, SB, JEM, MV, VEP]). The data analysis was also supported by the Bill & Melinda Gates Foundation (INV-030857 [JW, JEM, MV, VEP]). The funders had no role in study design, data collection and analysis, decision to publish, or preparation of the manuscript.

**Competing interests:** I have read the journal's policy and the authors of this manuscript have the following competing interests: AJP was the chair of the UK Department of Health and Social Care's Joint Committee on Vaccination until 2025, was a member of the WHO Strategic Advisory Group of Experts (SAGE) until 2022, and is the chair of the WHO Technical Advisory Group on Salmonella vaccines. VEP and FQ are members of the WHO SAGE typhoid working group. JDC and MAG are members of the WHO Technical Advisory Group on Salmonella vaccines. AJP receives grants from the Wellcome Trust, the Coalition for Epidemic Preparedness Innovations, Medical Research Council, National Institute for Health and Care Research, AstraZeneca, European Commission, and Serum Institute of India. VEP receives grants from Gavi, the Vaccine Alliance, US Centers for Disease Control and Prevention, National Institutes of Health/National Institute of Allergy and Infectious Diseases, and National Institute for Health and Care Research. All other authors declare no competing interests.

mixture model to the change in IgG concentration between participants' samples. We estimated the seroincidence of infection in a Bayesian framework for each study site, age group, and antigen target. Finally, we compared the seroincidence estimates with crude and adjusted estimates of clinical incidence based on blood-culture surveillance. Seroincidence estimates were significantly higher than enteric fever incidence across all study sites, age groups, and antigen targets, even after adjusting for underreporting (median ratio: 24.2, interquartile range: 11.4-58.9). Seroincidence consistently peaked in the 0–4-year age group and declined moderately between children and adults (33% to 58% decline in HlyE seroincidence between the 5–9 and 30 + year old age groups), while enteric fever incidence peaked in older children and fell sharply in adults (71% to 95% decline in adjusted clinical incidence). Seroincidence estimates based on the FliC, YncE, and HlyE antigens individually had the strongest correlation with observed enteric fever incidence across age groups and study sites ($r = 0.72$, $0.69$, and $0.63$, respectively). These findings suggest that in endemic settings, both children and adults are frequently infected by typhoidal *Salmonella* serotypes, although only a fraction of these infections present as clinically identifiable enteric fever cases.

## Author summary

Typhoid and paratyphoid fever cause over 10 million cases and 100,000 deaths each year. However, a lack of routine clinical surveillance data makes evidence-based public health decision making difficult. We collected paired blood samples several months apart from healthy children and adults in Bangladesh, Malawi, and Nepal, and tested these samples for Immunoglobulin G (IgG) antibodies against 7 typhoidal antigens. We then fit mixture models to the standardized change in IgG between the samples to identify infected participants, and used this data to estimate the age-specific seroincidence of infection for each antigen target and study site. Seroincidence estimates were significantly higher and peaked at an earlier age than enteric fever incidence, and remained substantial in adults. Seroincidence based on antibody responses to the FliC, YncE, and HlyE antigens were most strongly correlated with enteric fever incidence across study sites and age groups. Our findings show that the careful collection and analysis of serological data can provide valuable insights into the local transmission patterns and disease burden of enteric fever.

## Introduction

Enteric fever is an invasive bacterial disease caused by *Salmonella enterica* serotypes Typhi (*S.* Typhi) and Paratyphi A, B, or C (*S.* Paratyphi). Typhoidal *Salmonella* are spread through fecal-oral transmission, primarily via contaminated food and water

[1]. As a result, enteric fever mostly occurs in settings with limited access to clean water and sanitation in Africa and Asia, where it causes over 10 million cases and 100,000 deaths annually [2,3]. While paratyphoid fever is rare in Africa, it often occurs in the same regions as typhoid fever in Asia, although usually at a lower incidence [3,4].

Existing diagnostic tests for enteric fever have several limitations affecting their utility in endemic settings [5,6]. The Widal agglutination test detects antibodies against O and H antigens, and is simple and inexpensive to perform, but has relatively low sensitivity and specificity and requires paired samples for accurate interpretation [5,7]. Blood culture testing is highly specific but only ~60% sensitive, and requires several days to obtain a diagnosis [5,6,8]. Bone marrow culture is ~90% sensitive, but its invasiveness limits its scalability [5,8,9]. Testing blood with a polymerase chain reaction (PCR) assay after a brief culture period offers improved sensitivity and speed over blood culture testing alone [5,9,10]; however, both culture and PCR diagnostic tests require specialized training and equipment, which is not always available in endemic settings. Due to the lack of an affordable, accurate, and scalable diagnostic test, routine enteric fever surveillance is rarely performed [2,3,11]. The absence of robust and routine surveillance data limits policymakers ability to make evidence-based decisions around typhoid control and prevention [11].

In settings where routine clinical surveillance data is limited or unavailable, serosurveillance can provide valuable insights into local epidemiological conditions. Serosurveillance involves the collection and analysis of representative serologic data for the purpose of understanding population-level patterns of infection and immunity, and usually seeks to estimate either the incidence of infection (seroincidence) or the proportion of the population which has been infected (seroprevalence) [7,12]. Serosurveillance has been used to characterize the frequency and natural history of infection, identify risk factors, evaluate the risk of future outbreaks, and guide public health policy for a variety of infectious diseases, including measles [13], rubella [14], polio [15,16], pertussis [17,18], COVID-19 [19,20], and arboviruses [21,22].

Serologic studies of typhoidal *Salmonella* have most commonly used immunoglobulin G (IgG) against the Vi polysaccharide capsule produced by *S.* Typhi. However, anti-Vi IgG is often not elevated following infection, and the use of this antigen as the basis for typhoid conjugate vaccines (TCV) will confound its use as a marker of recent infection following vaccine introduction [7,23,24]. Spurred by the need for improved diagnostic tests, high-throughput immunoscreening of enteric fever cases has identified additional antigen targets that may be promising indicators of infection [7,23,25–27]. In particular, longitudinal studies following culture-confirmed enteric fever cases have shown that IgG antibodies against the haemolysin E (HlyE) and lipopolysaccharide (LPS) antigens rise sharply and remain significantly elevated for months following *S.* Typhi and *S.* Paratyphi A infection [23,25,28,29]. By modeling the kinetics of these antibodies and comparing them to cross-sectional measurements from the general population, Aiemjoy et al. estimated seroincidence in multiple typhoid-endemic settings [7,25,30].

The Strategic Typhoid Alliance Across Asia and Africa (STRATAA) study was conducted at sites in Bangladesh, Malawi, and Nepal to improve our understanding of typhoid transmission and epidemiology across disparate endemic settings [4,31]. Alongside enhanced clinical surveillance, a detailed household census, and healthcare utilization surveys, the STRATAA research consortium collected paired serological specimens from an age-stratified random sample of healthy participants at each study site. The incidence of seroconversion against the Vi antigen, defined as a two-fold rise in IgG on an enzyme-linked immunosorbent assay (ELISA), was higher than the adjusted incidence of typhoid fever and comparable between children and adults [4].

In this study, we use a novel multiplex assay to measure IgG responses against seven typhoidal antigens in paired blood samples from healthy children and adults in the Bangladesh, Malawi, and Nepal STRATAA sites. We then fit mixture models to data on the change in IgG over time to identify infected participants and estimate seroincidence. Finally, we compare the resulting seroincidence estimates to enteric fever incidence. This approach allows us to overcome the limitations of Vi-based serology and provide a comprehensive overview of typhoidal seroincidence, its variation across different age groups and settings, and its relationship with clinical disease in the STRATAA study.

## Methods

### Ethics statement

For clinical surveillance and the serosurvey, individual written informed consent was obtained from participants over the age of 18 or from a parent/guardian for participants below this age with additional assent sought from those between 11 and <18 years old. For the healthcare utilization survey (used to adjust enteric fever incidence for underreporting) and the household demographic census, written informed consent was obtained from the head of each household (as the 'key informant') on behalf of the entire household. The STRATAA study protocol received ethics approval from the Oxford Tropical Research Ethics Committee [39–15], the Malawi National Health Sciences Research Committee (15/5/1599), the University of Malawi College of Medicine Research Ethics Committee, the Nepal Health Research Council (306/2015) and the International Centre for Diarrhoeal Disease Research, Bangladesh Institutional Review Board (PR-15199).

**Study design and enrollment of participants.** The STRATAA study was performed in three urban communities: Mirpur thana in Dhaka, Bangladesh; Ndirande township in Blantyre, Malawi; and Lalitpur city in Kathmandu, Nepal. Among other factors, these areas differ in terms of demographics, population density, disease burden, and co-circulating pathogens; a detailed description of the sites is given in Darton et al.[31]. The baseline population of each site's demarcated study area was enumerated in a household census conducted between June and October 2016 [4]. A final census was performed at each site two years later, with two intermediate census updates at the Bangladesh site and a single update at the Nepal site. No census update was performed at the Malawi site.

At each site, the baseline census was used to randomly select residents for the serologic survey within five age groups: 0–4, 5–9, 10–14, 15–29, and 30+years old. The 0–4 and 5–9 year-old age groups were oversampled, relative to their share of the census population, to ensure that there would be enough participants to reliably estimate seroincidence in these age groups (S1 Fig) [31]. Each sampled resident was randomly assigned to a 3-month window within the year for sampling to ensure representative coverage over the study period. Field workers visited these residents at their home during this window to enroll participants and collect baseline blood samples. If the sampled resident was not available, the field workers attempted to enroll a different household member from the same age group. During this visit, field workers also attempted to enroll household members under 6 months of age bolster the number of participants from this age group.

Participants were enrolled on an ongoing basis between February 2017 and February 2018 in Bangladesh, between December 2016 and April 2018 in Malawi, and between January 2017 and May 2018 in Nepal. However, almost no children at the Nepal site were enrolled in the serosurvey after November 2017, due to participation in the TyVAC-Nepal TCV trial. Adults were not eligible for this trial and therefore continued to be enrolled in the serosurvey through the first half of 2018. To avoid confounding of age-specific incidence by calendar time, we excluded all participants at the Nepal site who were enrolled after November 2017. In addition, seroincidence estimates for Malawi could be biased due to seasonality in enteric fever incidence, since the Malawi serosurvey period was approximately 18 months and overlapped with two seasonal peaks in transmission. Therefore, we performed a sensitivity analysis for the Malawi site in which participants who did not have both samples collected during 2017 were excluded.

**Sample collection and laboratory testing.** Blood samples (1–3 ml) were collected from enrolled participants in ethylenediaminetetraacetic acid (EDTA) tubes. All three sites performed anti-Vi IgG ELISAs using VaccZyme kits from The Binding Site Group (see Meiring et al. for results [4]). A further aliquot of blood was sent to either the University of Cambridge or Oxford University for standardized IgG testing with a custom multiplex bead-based immunoassay. Carboxylated fluorescent beads (Luminex-Diasorin) were coupled to purified S. Typhi antigens and incubated with the heat-inactivated sera at 1:20 dilution in assay diluent (phosphate-buffered saline, 0.05% Tween-20 and 1% BSA) for 1 hour in the dark on a shaker. The antigen panel included cytolethal distending toxin B (CdtB), flagellin (FliC), HlyE, LPSO2, LPSO9, Vi, and YncE. IgG specific antibodies were detected with phycoerythrin (PE)-conjugated goat anti-human

IgG (Moss Inc) at 4µg/mL, incubated for 30 minutes in the dark on a shaker. Beads were washed and acquired on a Luminex 200 instrument.

To ensure comparability, paired samples from the same participant were always tested on the same plate using the same set of antigen batches. Each plate also included a blank control (diluent with no sera) and a negative control sample taken at baseline from a presumably unexposed participant in a typhoid human challenge study [32]. We tested each sample in duplicate and recorded the mean and standard deviation of the two fluorescence intensity (FI) measurements (after subtracting the associated blank FI value) for each antigen target. To quantify measurement variability, we calculated the coefficient of variation (CV) for each participant and antigen target as the ratio of the standard deviation to the mean of the duplicate FI measurements. Across participants, the median CV ranged from 1.56% (interquartile range (IQR): 0.64% to 3.43%) for HlyE at the Bangladesh study site to 9.40% (IQR: 4.07% to 18.99%) for Vi at the Nepal study site (S2 Fig).

**Data cleaning and processing.** Per the study protocol, we planned to collect a baseline and follow-up sample from each participant at two visits approximately 90 days apart. In practice, some participants were lost to follow-up, while others had their second sample collected earlier or later than the 90-day post-baseline target. To minimize the impact of false-negative infections, which could occur if antibodies have enough time to undergo significant waning by the time the second sample is collected, we excluded participants with over 150 days between samples; we conducted a sensitivity analysis in which this threshold was not applied, and another in which it was reduced to 100 days. The distribution of days between samples for each study site is shown in S3 Fig. We also excluded IgG measurements that had a non-positive FI value (indicative of measurement error), did not form a unique pair of baseline and follow-up measurements, or could not be linked to participant metadata.

To standardize measurements between antigen batches, we took the mean FI value from the two sample duplicates, applied a log10 transformation, and calculated a z-score based on the batch-specific mean and standard deviation of the resulting log10(FI) values. We refer to the resulting output as the standardized fluorescence intensity (sFI).

**Classifying seroresponses.** We fit a mixture of two linear regression models in which each qualifying participant is treated as a single observation, sFI at the first visit is the predictor variable, and the change in sFI between visits is the response variable. The fitted mixture model assigns each participant a posterior probability of membership in two clusters - one corresponding to a significant change (either increase or decrease) in IgG between visits, and the other corresponding to stable IgG between visits. We assigned each participant to the cluster with the higher posterior probability, and classified participants as infected if they had both a significant increase in IgG between visits (i.e., posterior probability of increase > 0.5) and a higher IgG concentration at the follow-up visit than a negative sample from the same plate, to exclude false-positive infections caused by small absolute rises in IgG around the assay limit of detection. All other participants were considered uninfected. This process is illustrated in Fig 1 for the HlyE antigen target. This mixture model approach takes advantage of the paired structure of the STRATAA serosurvey data, puts precise bounds on the time period during which exposure could have occurred, and does not require us to make specific assumptions about antibody kinetic parameters, the duration of post-infection immunity, or a seropositivity threshold.

We fit the mixture model separately for each of the seven antigen targets and three study sites, for a total of 21 fitted models. The full distribution of participants' posterior probability of infection from the mixture model are given in S4 Fig for each study site and target antigen. To test the convergence of our mixture models, we refit each model five times under different starting conditions. For each refitted mixture model, the number of participants classified as infected was within 2 of the model used in the main analysis (<0.15% of participants).

As a sensitivity analysis, we directly used each participant's mixture-model-derived probability of infection to estimate seroincidence based on a modified likelihood equation, described below, rather than classifying participants as infected or not using the cutoff value of 0.5. We also performed a sensitivity analysis in which all participants with a large relative

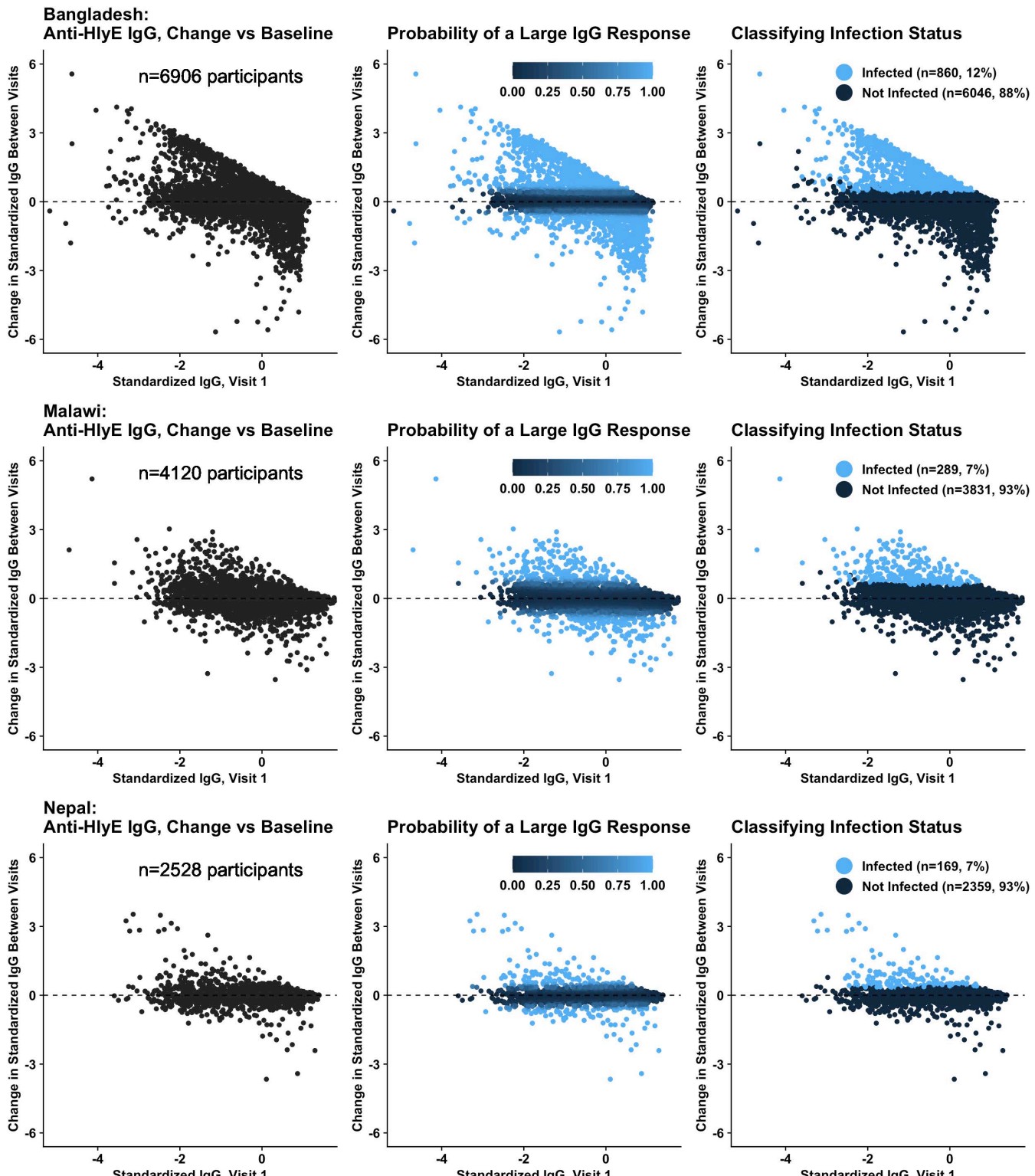

**Fig 1. Classification of Anti-HlyE IgG Responses.** IgG measurements refer to standardized log-transformed fluorescence intensity (see methods). Panels in the top, middle, and bottom rows correspond to the Bangladesh, Malawi, and Nepal study sites, respectively. In each panel, each point represents the anti-HlyE IgG at baseline (x-axis) and the change in IgG from baseline to the follow-up visit (y-axis) of a single participant. *Left*: Most

participants are clustered around the horizontal dashed line corresponding to no change in IgG between visits. *Middle*: Participants are colored by the posterior probability of having a large change (either an increase or decrease) in IgG between visits. This metric is derived from a two-cluster linear regression mixture model. *Right*: Participants who experienced a large rise in IgG between visits (posterior probability > 0.5) and had a higher IgG level at follow-up than a matched negative control were classified as infected during this period. All other participants were considered to be uninfected.

increase in IgG (posterior probability of increase > 0.5) are considered infected, even if the absolute concentration of IgG in the follow-up sample is low.

**Estimating seroincidence.** We assumed that participants experience new typhoidal *Salmonella* infections at a mean rate $\lambda$ (seroincidence), such that the time $t$ from visit 1 to the next infection is exponentially distributed. The likelihood function for $\lambda$ is formed by linking infection status to the cumulative distribution function (CDF) of the exponential distribution:

$$L\left(\lambda\middle|I\right) = \sum_{i=1}^{n} I_i P(t < d_i) + (1 - I_i)P(t > d_i) = \sum_{i=1}^{n} I_i(1 - e^{-\lambda d_i}) + (1 - I_i)e^{-\lambda d_i},$$

where $i = 1, 2, ..n$ denotes the set of participants, $I_i$ is a binary variable indicating whether or not participant $i$ was infected between samples ($I_i$ = 1 or 0, respectively), and $d_i$ is the time interval between samples. We implemented this likelihood function in a Markov chain Monte Carlo (MCMC) framework with the Metropolis-Hastings [33] algorithm to estimate the posterior distribution of $\lambda$, using an approximately flat gamma prior (shape = scale = 0.001), three independent chains of 10,000 samples, a 1,000 sample burn-in period, and a normal proposal distribution with a variance of 0.1 infections per person-year. We visually examined the output chains to confirm proper convergence. Seroincidence was estimated separately for each of the three study sites, seven antigen targets, and five age groups. Finally, we calculated the "overall" age-standardized seroincidence for each study site and antigen as a census-weighted average of the age-specific posterior distributions of $\lambda$. As a sensitivity analysis, we replaced $I_i$ in the likelihood function with the posterior probability that participant $i$ was infected, in order to account for uncertainty in the mixture model's classification of infection status.

**Clinical surveillance.** In order to evaluate the relationship between serologically-inferred infections and symptomatic disease at the population level, we compared our seroincidence estimates to the incidence of blood-culture-confirmed enteric fever (typhoid and paratyphoid fever) at each study site. In the STRATAA study, members of the household census population who presented to a study healthcare facility with either a recorded temperature of 38°C or a history of fever for ≥48 hours were approached for recruitment, and consenting patients provided blood for culture testing [4]. At each study site, clinical surveillance activities and recruitment for the serologic survey began within a month of each other (Table 1). For Bangladesh and Malawi, we used the full ~2-year clinical surveillance period to calculate enteric fever incidence, as this almost entirely overlaps with the interval covered by the serosurvey (Table 1). For Nepal, we only used the first year of clinical surveillance (2017) to calculate enteric fever incidence in order to align with the period covered by our serologic samples (see above). In addition to crude incidence, we also present the symptomatic incidence of enteric fever, which has been adjusted for blood culture test sensitivity and the probability of care-seeking and testing [4].

All modeling and analysis was performed in R (v3.6.3) [34]. The mixture model was fitted using the *flexmix* function of the *mixtools* R package (v1.2.0), which employs an expectation-maximization algorithm [35]. Code and data are available at *github.com/pitzerlab/Typhoid-Seroincidence*.

## Results

The number of participants meeting the inclusion criteria (collection of both baseline and follow-up samples within 150 days of each other, a complete record of the sample collection dates, and enrollment before December 2017 at the Nepal

**Table 1.  Participant Number and Characteristics.**

|  | Bangladesh | Malawi | Nepal |
|---|---|---|---|
| Serosurvey Enrollment Period | Feb 2017-Feb 2018 | Dec 2016-Apr 2018 | Jan-Nov 2017 |
| Clinical Surveillance Period | Jan 2017-Dec 2018 | Nov 2016-Oct 2018 | Jan-Dec 2017 |
| Total Participants | 6,967 | 4,231 | 2,557 |
| Excluded Participants: | 115 | 578 | 8 |
| Measurement error | 11 | 0 | 0 |
| No follow-up visit | 39 | 38 | 1 |
| Indeterminate visit date | 3 | 70 | 6 |
| >150 days between visits | 62 | 470 | 1 |
| Participants Included in Final Analysis | 6,852 | 3,653 | 2,549 |
| Participants by Age (%) |  |  |  |
| 0-4 years | 1,914 (27.9%) | 704 (19.3%) | 194 (7.6%) |
| <6 months | 16 (0.2%) | 2 (<0.1%) | 0 |
| 6 months-<1 year | 61 (0.9%) | 17 (0.5%) | 4 (0.2%) |
| 1-4 years | 1,837 (26.8%) | 685 (18.8%) | 190 (7.4%) |
| 5-9 years | 1,115 (16.3%) | 663 (18.1%) | 362 (14.2%) |
| 10-14 years | 702 (10.2%) | 417 (11.4%) | 315 (12.4%) |
| 15-29 years | 955 (13.9%) | 655 (17.9%) | 345 (13.5%) |
| 30+years | 2,166 (31.6%) | 1,214 (33.2%) | 1,333 (52.3%) |
| Days Between Baseline and Follow-Up Samples: Median (IQR) | 84 (79 - 89) | 103 (91 - 113) | 110 (101-122) |

site to avoid overlap with the TyVac-Nepal TCV trial) was 6,852 at the Bangladesh site, 3,653 at the Malawi site, and 2,549 at the Nepal site (Table 1). Follow-up samples were collected slightly earlier in Bangladesh than in Malawi and Nepal, with a median of 84, 103, and 110 days between visits, respectively (S3 Fig). Baseline IgG levels against each target antigen were generally stable or slightly increased with age (S5 Fig). Standardized IgG measurements for different antigens from the same participant sample were almost always positively correlated with each other, although most of the correlations were weak to moderate: Pearson's correlation coefficients for antigen pairs ranged from -0.05 to 0.59 at the Bangladesh site, from 0.1 to 0.80 at the Malawi site, and from 0.17 to 0.75 at the Nepal site (S6 Fig).

There were 47 participants who received a blood-culture test from a STRATAA study clinic during the period between the collection of the first and second serologic samples (37 from Bangladesh, 10 from Nepal, none from Malawi) (S7 Fig). Four participants were blood-culture positive for either S. Typhi (n=3) or S. Paratyphi A (n=1) and experienced a 4.0-fold median rise in anti-HlyE IgG (IQR: 3.35-fold to 4.60-fold). One blood-culture-negative participant at the Bangladesh site had a positive stool-culture for S. Typhi, and experienced a 4.76-fold rise in HlyE IgG. All five of these participants were classified as infected by the HlyE mixture model. The 42 participants without a positive S. Typhi/Paratyphi culture test experienced only a 1.01-fold median change in anti-HlyE IgG (IQR: 0.89-fold to 1.10-fold), although 7 of these culture-negative participants were classified as infected by the HlyE mixture model.

Seroincidence was considerably greater than enteric fever incidence across age groups, antigen targets, and study sites, even after adjusting for underreporting (Fig 2, median ratio: 24.2, IQR: 11.4-58.9). For instance, median age-standardized seroincidence estimates based on the HlyE antigen were 46.3 (95% credible interval (CrI): 39.3 to 54.2) per 100 person-years (PY) in Bangladesh, 24.0 (95% CrI: 17.9 to 31.5) per 100 PY in Malawi, and 21.0 (95% CrI: 14.6 to 29.2) per 100 PY in Nepal (Table 2). These seroincidence estimates are 32, 54, and 18 times greater than the adjusted enteric fever incidence and 227, 415, and 265 times greater than the unadjusted enteric fever incidence at the Bangladesh, Malawi, and Nepal study sites, respectively.

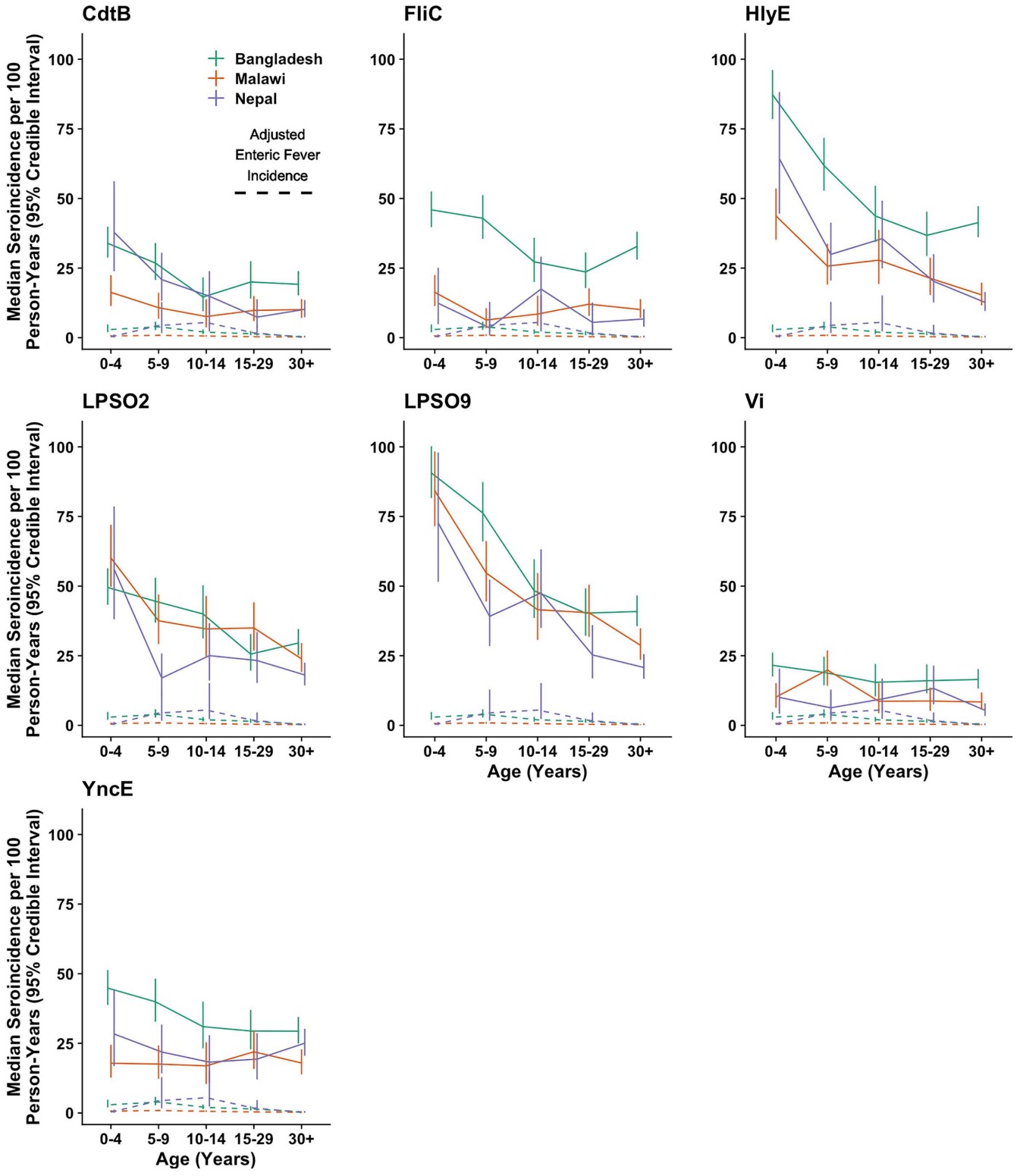

**Fig 2. Seroincidence by Age, Antigen, and Study Site.** Each panel corresponds to the specific antigen target which was used to classify participants' infection status when calculating seroincidence. Solid lines denote the median seroincidence (y-axis) in each age group (x-axis). Vertical lines represent the 95% credible intervals of the seroincidence estimates, and dashed lines indicate the adjusted incidence of enteric fever. Green, orange, and purple lines correspond to the Bangladesh, Malawi, and Nepal study sites, respectively.

Table 2. HlyE Seroincidence and Clinical Enteric Fever Incidence by Age and Study Site.

| Age Group (Years) | Dhaka, Bangladesh | | | | Blantyre, Malawi | | | | Kathmandu, Nepal | | | |
|---|---|---|---|---|---|---|---|---|---|---|---|---|
| | Unadjusted Enteric Fever Cases per 100 PY (95% CI) | Adjusted Enteric Fever Cases per 100 PY (95% CrI) | Median HlyE Sero-incidence per 100 PY (95% CrI) | Ratio of Sero-incidence to Adjusted (Unadjusted) Enteric Fever Incidence | Unadjusted Enteric Fever Cases per 100 PY (95% CI) | Adjusted Enteric Fever Cases per 100 PY (95% CrI) | Median HlyE Sero-incidence per 100 PY (95% CrI) | Ratio of Sero-incidence to Adjusted (Unadjusted) Enteric Fever Incidence | Unadjusted Enteric Fever Cases per 100 PY (95% CI) | Adjusted Enteric Fever Cases per 100 PY (95% CrI) | Median HlyE Sero-incidence per 100 PY (95% CrI) | Ratio of Sero-incidence to Adjusted (Unadjusted) Enteric Fever Incidence |
| 0-4 | 0.47 (0.38-0.58) | 2.96 (1.99-4.79) | 87.29 (78.5-96.2) | 29 (186) | 0.08 (0.05-0.12) | 0.63 (0.40-0.96) | 43.73 (35.15-53.59) | 69 (527) | 0.03 (0.004-0.11) | 0.30 (0.04-1.20) | 64.23 (44.56-41.32) | 212 (2018) |
| 5-9 | 0.67 (0.55-0.81) | 3.91 (2.76-5.77) | 61.83 (52.80-71.78) | 16 (92) | 0.15 (0.10-0.20) | 0.86 (0.60-1.20) | 25.71 (19.09-33.74) | 30 (176) | 0.22 (0.11-0.39) | 4.38 (1.67-12.87) | 29.92 (20.55-41.32) | 7 (134) |
| 10-14 | 0.34 (0.26-0.44) | 1.98 (1.33-3.01) | 43.71 (34.42-54.57) | 22 (129) | 0.09 (0.06-0.13) | 0.60 (0.38-0.92) | 27.86 (19.34-38.67) | 46 (317) | 0.28 (0.17-0.43) | 5.47 (2.27-15.17) | 35.57 (24.89-49.22) | 7 (128) |
| 15-29 | 0.14 (0.11-0.18) | 1.41 (0.89-2.41) | 36.76 (29.36-45.27) | 26 (255) | 0.03 (0.02-0.05) | 0.36 (0.22-0.57) | 21.35 (15.36-28.71) | 59 (667) | 0.11 (0.08-0.16) | 1.75 (0.78-4.63) | 20.27 (12.73-29.97) | 12 (181) |
| 30+ | 0.04 (0.03-0.06) | 0.40 (0.23-0.74) | 41.32 (36.10-47.24) | 103 (990) | 0.02 (0.01-0.04) | 0.25 (0.12-0.45) | 15.40 (11.62-19.79) | 62 (733) | 0.01 (0.005-0.03) | 0.22 (0.07-0.69) | 12.69 (9.58-16.42) | 59 (882) |
| Overall | 0.20 (0.18-0.23) | 1.44 (1.14-1.48) | 46.31 (39.29-54.19) | 32 (227) | 0.06 (0.05-0.07) | 0.44 (0.35-0.72) | 24.05 (17.91-31.46) | 54 (415) | 0.08 (0.06-0.10) | 1.15 (0.71-1.98) | 21.09 (14.56-29.23) | 18 (265) |

Seroincidence was generally highest in the 0–4-year age group and declined gradually with age, but remained high even in adults (Fig 2). Exceptions include seroincidence based on the FliC antigen in Malawi and Nepal, seroincidence based on the YncE antigen in Malawi, and seroincidence based on the Vi antigen at each study site, which were approximately stable with age. HlyE seroincidence was particularly high in the first two years of life at the Bangladesh and Malawi study sites, and among 2–3-year-olds at the Nepal site (Fig 3). Excluding participants who were younger than 6 months old at enroll-ment (16 at the Bangladesh site, two at the Malawi site, and none at the Nepal site) resulted in similar or slightly higher HlyE seroincidence estimates (S8 Fig). The observed relationship between seroincidence and age contrasts with enteric fever incidence, which peaks in school-aged children before rapidly declining to a low level in adults (Fig 4). For instance, adjusted enteric fever incidence declined 90% at the Bangladesh site, 71% at the Malawi site, and 95% at the Nepal site between the 5–9 and 30＋year age groups, while HlyE seroincidence declined only 33%, 40%, and 58%, respectively, over the same interval.

Seroincidence estimates were positively correlated with the unadjusted incidence of enteric fever across different age groups and study sites ($r$＝0.25 to 0.72 across antigen targets) (S9 Fig and Table 3). Adjusting the incidence of enteric

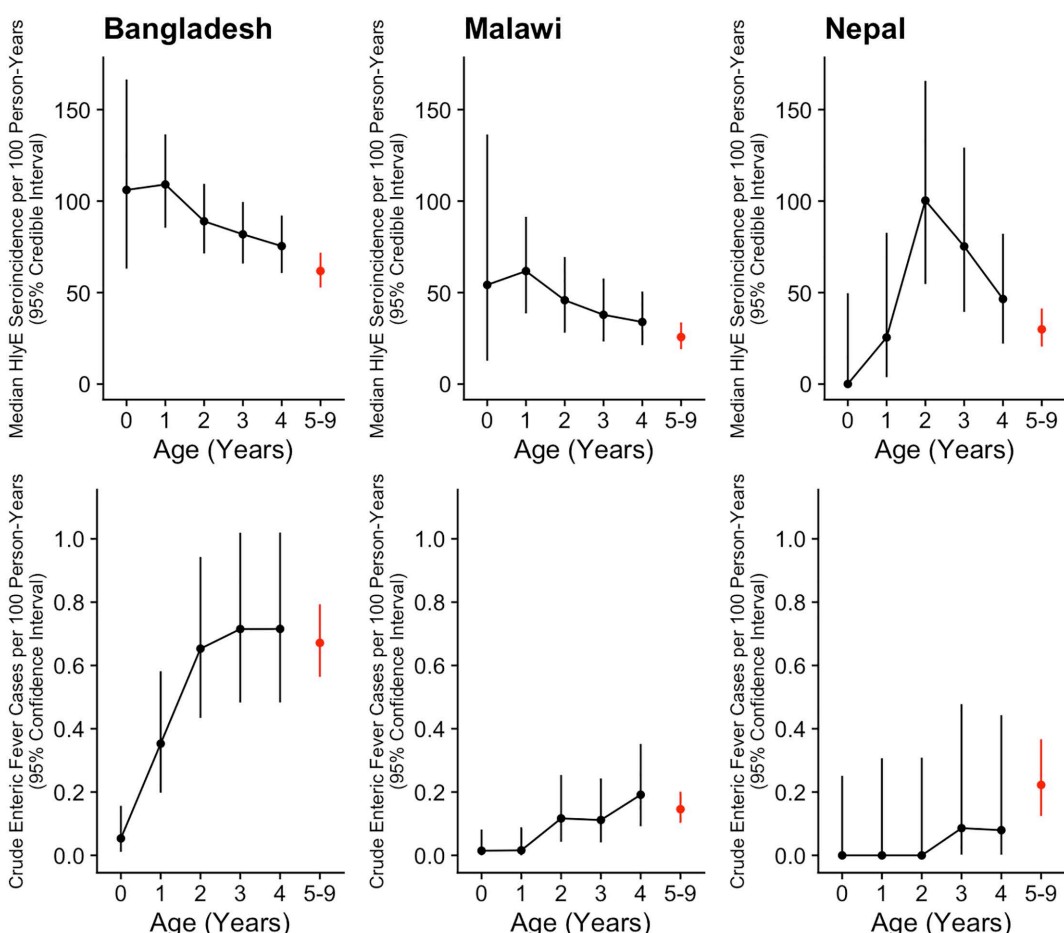

**Fig 3. HlyE Seroincidence and Enteric Fever Incidence by Age in Young Children.** Panels in the top and bottom rows display HlyE seroincidence and crude enteric fever incidence, respectively, at the Bangladesh (left), Malawi (middle), and Nepal (right) study sites. Age-specific incidence is shown for each of the first 5 years of life (black), and for 5-9 year-olds overall (red). Vertical lines represent 95% credible and confidence intervals for seroinci-dence and enteric fever incidence, respectively.

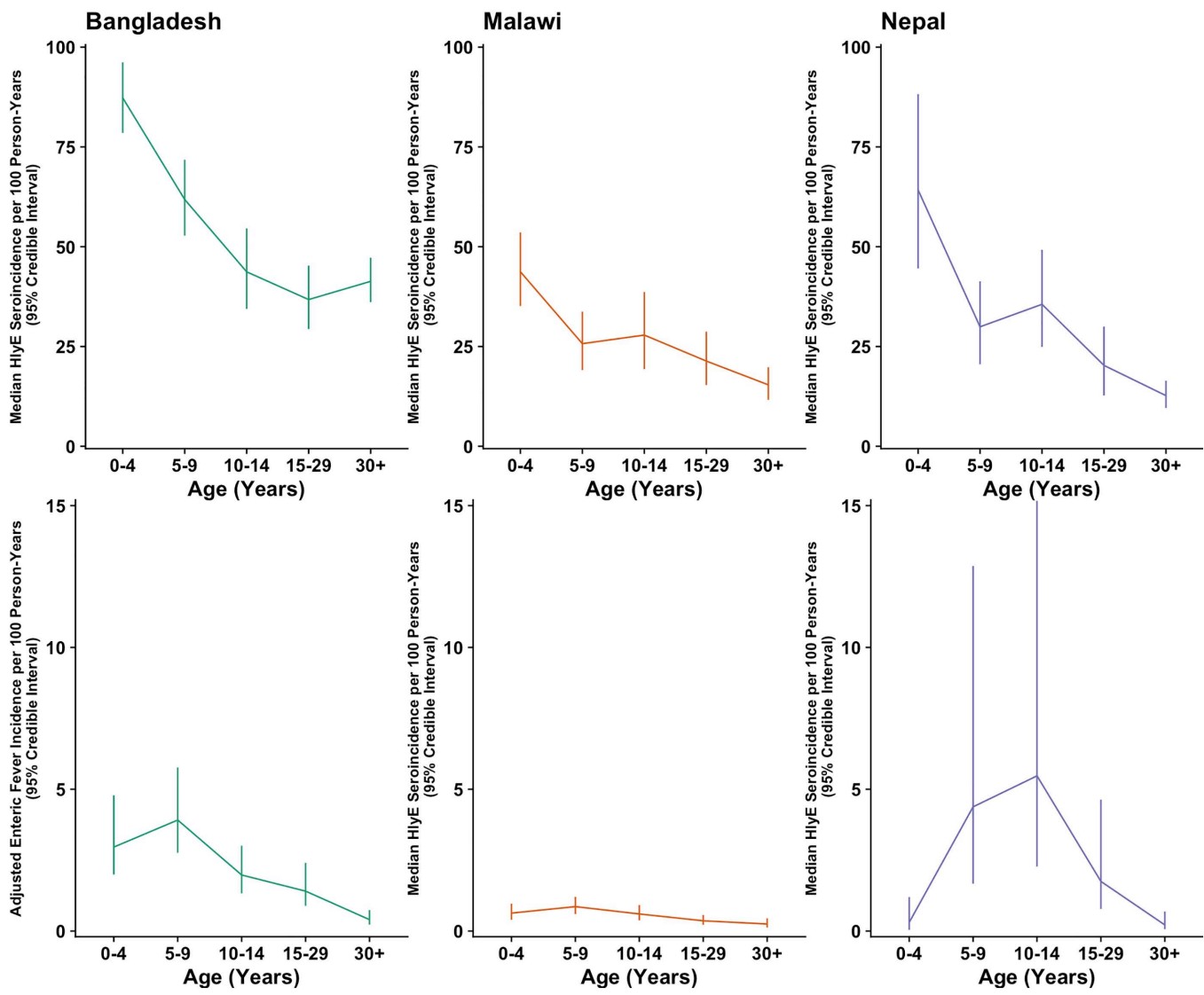

**Fig 4. Relative Trend in HlyE Seroincidence and Enteric Fever Incidence by Age and Study Site.** Each column of corresponds to a different study site. Panels in the top row display the median seroincidence based on the HlyE antigen for each age group, while the panels in the bottom row display the adjusted incidence of enteric fever cases for each age group. Seroincidence is highest in the 0-4-year age group and declines relatively gradually with age, while the incidence of enteric fever peaks in older children before rapidly declining to a low level in adults. Vertical lines represent the 95% credible intervals of seroincidence estimates.

fever for underreporting weakened its correlation with seroincidence ($r=-0.16$ to $0.31$, S10 Fig), although this appears to be attributable to the adjustment factors for Nepal: when estimates from the Nepal study site were excluded, the correlation between seroincidence and enteric fever incidence increased for all antigens, and was comparable between unadjusted ($r=0.43$ to $0.82$) and adjusted enteric fever incidence ($r=0.40$ to $0.86$) (Table 3). Enteric fever incidence was most strongly correlated with seroincidence estimates based on the FliC, YncE, HlyE, and CdtB (for comparisons with adjusted incidence only) antigen targets. YncE and CdtB were the only antigen targets for which the median seroincidence estimates captured the overall rank ordering of enteric fever incidence between study sites, with the highest incidence at the Bangladesh site, the lowest incidence at the Malawi site, and intermediate incidence at the Nepal site. However, the

**Table 3. Correlation Between Seroincidence and Enteric Fever Incidence Across Age Groups and Study Sites.**

| Antigen Target | Pearson correlation between seroincidence* and unadjusted enteric fever incidence | | Pearson correlation between seroincidence* and adjusted enteric fever incidence | |
|---|---|---|---|---|
| | Bangladesh, Malawi, and Nepal | Bangladesh and Malawi only | Bangladesh, Malawi, and Nepal | Bangladesh and Malawi only |
| CdtB | 0.44 | 0.74 | 0.26 | 0.79 |
| FliC | 0.72 | 0.78 | 0.30 | 0.81 |
| HlyE | 0.63 | 0.78 | 0.31 | 0.81 |
| LPSO2 | 0.25 | 0.43 | -0.16 | 0.40 |
| LPSO9 | 0.54 | 0.64 | 0.21 | 0.64 |
| Vi | 0.59 | 0.67 | 0.15 | 0.69 |
| YncE | 0.69 | 0.82 | 0.25 | 0.86 |

*median age-specific seroincidence estimate for the indicated antigen target.

95% credible intervals of the age-standardized YncE and CdtB seroincidence estimates overlapped between study sites (S11 Fig). Limiting our Malawi analysis to participants with both samples collected in 2017 resulted in lower seroincidence estimates for each antigen target (S11 and S12 Figs). As a result, the rank ordering of study sites by age-standardized seroincidence matched that of enteric fever incidence for the CdtB, HlyE, Vi, and YncE antigen targets in this sensitivity analysis. While HlyE seroincidence in children was higher at the Nepal study site than at the Malawi site, seroincidence in adults was comparable between the two sites.

To evaluate temporal trends, we assigned each serosurvey participant to the month containing the midpoint between sample collection dates and calculated HlyE seroincidence for each month and study site (S13 Fig). At the Bangladesh site, seroincidence tracked the rise in enteric fever cases and blood-culture-positivity over the study period. In Malawi, both seroincidence and enteric fever cases peaked simultaneously at the beginning of 2017 and 2018, during the rainy season. At the Nepal study site, a peak in seroincidence in the spring of 2017 coincided with low enteric fever cases, although this could be explained by the low number of patients enrolled in clinical surveillance in the early months of the STRATAA study [4]. Subsequently, seroincidence tracked the decline in enteric fever cases and blood-culture positivity following the summer rainy season (June to August), but did not capture their resurgence in the last two months of 2017.

Reducing the upper limit on time between samples from 150 to 100 days did not have a large or consistent impact on seroincidence estimates (S14 Fig). Removing this limit and including participants with over 150 days between samples had very little impact on seroincidence at the Bangladesh and Nepal study sites, since few participants were excluded by this criterium (S3 Fig). At the Malawi site, seroincidence estimates marginally decreased across age groups and antigen targets when the 470 participants with >150 days between samples were included in the analysis, although the relationship between age and seroincidence was essentially unchanged.

Classifying all participants with a large relative increase in IgG between visits as infected, regardless of the absolute IgG concentration at the second visit, increased our seroincidence estimates, although the effect varied between antigen targets (S15 Fig). Seroincidence based on the Vi antigen increased significantly at each of the study sites, particularly in the younger age groups. However, estimates based on the HlyE, LPSO9, and YncE antigens were mostly unchanged. With the exception of the Vi antigen, our decision to classify all participants with a low absolute IgG concentration at follow-up as uninfected did not meaningfully affect the relationship between age and seroincidence. Finally, seroincidence estimates were consistently lower (but still substantially higher than enteric fever incidence) when we replaced binary infection status with the posterior probability of infection from the mixture model in the likelihood function (S16 Fig).

## Discussion

In this study, we estimated the incidence of serologically-defined typhoidal *Salmonella* infection in three endemic urban settings by analyzing paired blood samples from the general population. These seroincidence estimates were substantially higher than the incidence of enteric fever across sites and age groups, even after adjusting for underreporting. This suggests that residents of endemic settings are infected by typhoidal *Salmonella* every few years on average, and that most of these infections are asymptomatic or do not present with traditional enteric fever symptoms. In addition to their difference in magnitude, seroincidence and enteric fever incidence followed different relative patterns with age: while seroincidence was highest in the first few years of life, enteric fever incidence was relatively low during this period and peaked later in childhood, as is typical in many typhoid-endemic settings [2,4,36–38]. Our finding of higher seroincidence in young children is consistent with other serologic studies [4,25], and contradicts the idea that enteric fever incidence is low in young children because they are not exposed as often as older children. Cases in young children may have been under-detected in the STRATAA study, relative to other age groups [39]. There is some evidence that the presentation of enteric fever is less specific, and potentially less severe, in young children, which could affect care-seeking, and parents of young children may have been less likely to consent to the collection of blood for culture diagnosis [1,40,41]. Blood samples from young children, particularly infants, are also more difficult to collect and lower in volume than those from older patients, reducing culture sensitivity [8]. While the adjustment factors applied to the culture-confirmed incidence data in the STRATAA study varied by age and attempted to control for some of these factors, the methodology may not have fully captured all of the differences between age groups. The low incidence of enteric fever in adults, on the other hand, likely reflects the gradual development of adaptive immunity from repeated exposure events throughout the lifespan [42]. Our seroincidence results reveal that the low burden of disease in adults only can be partially explained by a reduced risk of infection, which suggest that infections in this group are more likely to be asymptomatic or only result in mild disease. The relatively high seroincidence in adults suggests that they are frequently infected by typhoidal *Salmonella*, and therefore may be more important drivers of transmission than their disease burden alone would suggest; this may explain why only vaccinating children <15 years old failed to generate strong indirect protection in a cluster-randomized trial of TCV in Dhaka [43].

Seroincidence estimates based on the FliC, HlyE and YncE antigen targets were most strongly correlated with enteric fever incidence between age groups and settings. In a comparison of longitudinal antibody responses in culture-diagnosed enteric fever cases, a previous study showed that anti-HlyE IgG levels increased sharply following diagnosis and remained significantly elevated, relative to healthy controls from the community and culture-negative patients, for at least three months in both typhoid and paratyphoid A cases [23]. Anti-YncE IgG levels, which have primarily been studied as a potential marker of chronic *S.* Typhi carriage [44], also rose and remained elevated following the onset of enteric fever, but these responses were much less distinctive against the background level of antibodies in non-cases. FliC was not included in this comparison, although cross-reactivity with is known to be an issue for serodiagnostics based on flagellar antigens [5,45]. Due to their strong association with enteric fever cases at both the individual and population level, antibody responses against the HlyE antigen should be prioritized as an endpoint in future seroepidemiology studies.

Our seroincidence estimates were similar in magnitude to those obtained with an alternative approach leveraging cross-sectional serology data [25]. In both studies, overall seroincidence was > 5 per 100 person-years for each study site and antigen, and were substantially higher than the adjusted incidence of enteric fever in their respective populations. Both studies included estimates from sites in Dhaka and Kathmandu, and the ratio of overall seroincidence to adjusted enteric fever incidence was similar between the two studies in these locations (32 for both studies in Dhaka, and 18 vs 16 in Kathmandu), which suggests that both methods produce similar results. Since paired and cross-sectional samples resulted in similar seroincidence estimates, the logistical benefit of not having to collect a follow-up sample may make cross-sectional sampling the more attractive option for future seroepidemiology studies of typhoidal *Salmonella*. By

contrast, our Vi seroincidence estimates were higher than the previously-published ELISA-based Vi seroincidence estimates from the STRATAA study at the Bangladesh and Malawi sites (although estimates were similar at the Nepal site), despite being based on the same target antigen and participant samples (S17 Fig). The multiplex bead assay used in this study may have been able to measure IgG antibodies more precisely than the traditional ELISA method, which would have allowed us to detect serological exposures more easily.

Our analysis is subject to several limitations which should be considered when interpreting our results. First, we assumed that a large rise in IgG against the target antigen indicated typhoidal *Salmonella* infection, but this response could also arise from cross-reactive exposures to related bacteria. Cross-reactivity would inflate our seroincidence estimates and weaken their association with enteric fever incidence at the population level. For instance, the relatively high LPSO2/O9 seroincidence at the Malawi study site could be partially explained by greater exposure to non-typhoidal *Salmonella* strains in this setting, relative to Bangladesh and Nepal [4]. As described above, anti-HlyE IgG has a strong and specific association with recent enteric fever cases [23,25]; as a result, seroincidence estimates based on this target should be fairly robust to the impact of cross-reactive exposures. Second, our approach is premised on the idea that IgG will rise sharply and remain significantly elevated for several months following infection. This was demonstrated previously in a longitudinal study of typhoid and paratyphoid fever cases, but it is possible that the magnitude and shape of antibody responses varies between clinical and asymptomatic cases [23]. If asymptomatic infections only result in a small or transient rise in IgG antibodies, then some participants could have been misclassified as uninfected. Third, our methodology assumes that seroincidence is constant over time. In reality, incidence varies seasonally and from year-to-year, and therefore our estimates should be treated as averages over the study period. Incidence can also vary between subgroups of participants with different risk factors. We attempted to control for this by recruiting participants from a randomly selected sample of residents enumerated through the STRATAA household census. Fourth, there were minor differences in the timing of sample collection between the study sites. Follow-up samples were usually collected slightly earlier at the Bangladesh site relative to the Malawi and Nepal sites. However, in a sensitivity analysis that excluded participants with over 100 days between samples, in effect negating the difference between Bangladesh and the other two sites, we did not observe any systematic change in the seroincidence estimates. Fifth, very few participants under 6 months of age were enrolled in our study, despite efforts to recruit from this population. Thus, we were not able to reliably evaluate the effect of maternal immunity in this specific age group. Sixth, the estimated adjustment factors for enteric fever incidence were higher at the Nepal site than at the Bangladesh and Malawi study sites, particularly in the 5–9 and 10–14 year-old age groups. Overestimation of these adjustment factors at the Nepal site could have weakened the observed correlation between enteric fever incidence and seroincidence. Finally, with the exception of Vi (which is only present in *S.* Typhi), the target antigens in this study are expressed by both *S.* Typhi and *S.* Paratyphi. As a result, seroincidence estimates based on these antigens do not distinguish between the two serotypes. However *S.* Paratyphi is not believed to circulate at the Malawi study site, so exposures in this setting can be attributed to *S.* Typhi [4].

We conducted a seroepidemiologic study of typhoidal *Salmonella* infection dynamics at three study sites in Bangladesh, Malawi, and Nepal. We found that the seroincidence of infection was much higher than symptomatic enteric fever incidence, varied with age, and was predictive of the population-level disease burden, particularly for the HlyE, YncE, and FliC antigen targets. We recommend that future serosurveillance studies focus on the HlyE antigen target, given its strong association with enteric fever at the population and individual level. These findings strengthen our understanding of the transmission dynamics and natural history of *S.* Typhi and *S.* Paratyphi serotypes, and can inform the design of future serologic studies seeking to generate data on these pathogens. Given the high seroincidence observed among adults, future studies should evaluate the public health impact and cost-effectiveness of vaccinating adults against typhoid fever. Even if infections in this age group are relatively unlikely to progress to clinically severe disease, there could be indirect benefits on population-level transmission.

## Supporting information

**S1 Fig. Age Distribution of Serosurvey Participants and the Census Population.** The height of each bar indicates the proportion of serosurvey participants (light grey, left) and the baseline census population (dark grey, right) in each age group for each study site.
(PNG)

**S2 Fig. Coefficients of Variation of Duplicate Sample Measurements.** Boxplots summarize the distribution of participant's coefficient of variation (CV) values (y-axis) for duplicate fluorescence intensity (FI) measurements. Each box corresponds to a different antigen target (x-axis), while each panel corresponds to a different study site. Box whiskers extend from the minimum to the maximum CV values.
(PNG)

**S3 Fig. Distribution of Days Between Paired Serologic Samples.** Left: Boxplots of the time between the collection of baseline and follow-up samples for participants at the Bangladesh, Malawi, and Nepal study sites. Box whiskers extend from the minimum to the maximum value. Right: Histograms of the number of participants at each study site with 0–50, 51–100, 101–150, and >150 days between samples at the Nepal (top), Malawi (middle), and Bangladesh (bottom) study sites. *Participants with >150 days between samples were excluded from the primary analysis.
(PNG)

**S4 Fig. Distribution of the Mixture Model Posterior Probability of Infection Among Participants with a Rise in IgG.** Each panel is a histogram of the posterior probability of infection (as indicated by a large rise in IgG) for participants who experienced an increase in IgG between the baseline and follow-up visits. Participants who experienced a decrease in IgG between visits were assumed to have not been infected and are not included in this figure. Each column of panels corresponds to a different antigen target, while the top, middle, and bottom rows correspond to the Bangladesh, Malawi, and Nepal study sites.
(PNG)

**S5 Fig. Baseline IgG by Age and Study Site.** Each panel corresponds to a specific antigen target against which IgG antibodies were measured. Solid lines denote the median of the log10-transformed fluorescence intensity (FI, a proxy for IgG concentration, y-axis) across participant's baseline samples in each age group (x-axis). Vertical lines represent the interquartile range (IQR) of the log10(FI) measurements. Green, orange, and purple lines correspond to the Bangladesh, Malawi, and Nepal study sites, respectively.
(PNG)

**S6 Fig. Correlation of IgG Between Antigen Targets.** This figure displays the Pearson's correlation coefficient of standardized IgG measurements from the same participant sample for each unique pair of antigen targets.
(PNG)

**S7 Fig. Anti-HlyE IgG Responses in Participants with Blood-Culture Testing.** Each point represents a single serosurvey participant who also had a blood-culture test performed at a STRATAA study clinic during the period between their two serologic samples. The y-axis indicates the fold-change in anti-HlyE IgG between the baseline and follow-up sample. Participants classified as infected by the mixture model on the basis of their HlyE seroresponse are plotted as triangles. All other participants are plotted as circles. The color of each point indicates the blood-culture test result. Individual points have been adjusted horizontally to prevent visual overlap.
(PNG)

**S8 Fig. Effect of Including Participants Under Six Months of Age on HlyE Seroincidence Estimates.** Points and lines represent the median estimate and 95% credible interval of HlyE seroincidence, respectively. Panels in the top row

correspond to the Bangladesh study site, while those in the bottom row correspond to the Malawi site. Panels in the left column are based on participants who were <1 year old at the time of enrollment, while panels in the right column are based on participants who were <5 years old at enrollment. Within each panel, the values on the left and right represent the seroincidence estimate when participants <6 months old are included and excluded, respectively. No participants <6 months of age were enrolled at the Nepal site, so it is not included in this figure.
(PNG)

**S9 Fig. Correlation Between Seroincidence and Unadjusted Enteric Fever Incidence.** Each panel corresponds to the specific antigen target which was used to classify participants' infection status when calculating seroincidence. Each point corresponds to an age group (shape) at a given study site (color). The position of each point represents the seroincidence (x-axis) and unadjusted enteric fever incidence (y-axis) for that age group and study site during the study period. The linear relationship between seroincidence and enteric fever incidence is indicated by a dashed line of best fit and the Pearson's correlation coefficient (r) at the top of the panel.
(PNG)

**S10 Fig. Correlation Between Seroincidence and Adjusted Enteric Fever Incidence.** Each panel corresponds to the specific antigen target which was used to classify participants' infection status when calculating seroincidence. Each point corresponds to an age group (shape) at a given study site (color). The position of each point represents the seroincidence (x-axis) and adjusted enteric fever incidence (y-axis) for that age group and study site during the study period. The linear relationship between seroincidence and enteric fever incidence is indicated by a dashed line of best fit and the Pearson's correlation coefficient (r) at the top of the panel.
(PNG)

**S11 Fig. Overall Age-Standardized Seroincidence and Enteric Fever Incidence by Study Site.** Each bar corresponds to a different study site, and the bar heights represent either the seroincidence of infection or the incidence of enteric fever cases at each site, as indicated. Error bars represent the 95% credible interval (CrI) for the estimates of seroincidence and adjusted enteric fever incidence, and 95% confidence intervals (CI) for unadjusted enteric fever incidence. Incidence measures in this figure correspond to the overall population at each study site and its unique age distribution, rather than any specific age stratum. Since the Malawi serosurvey overlapped with two seasonal peaks in transmission, we included a sensitivity analysis limiting the Malawi site to participants with both samples collected in 2017.
(PNG)

**S12 Fig. Seroincidence at the Malawi Site by Age, Antigen, and Time Period.** Each panel corresponds to the specific antigen target which was used to classify participants' infection status when calculating seroincidence. Solid lines denote the median seroincidence (y-axis) in each age group (x-axis). Vertical lines represent the 95% credible intervals of the seroincidence estimates, and dashed lines indicate the adjusted incidence of enteric fever. Orange lines represent estimates based on the full serosurvey period, while green lines correspond to estimates limited to participants with both samples collected in 2017.
(PNG)

**S13 Fig. Monthly Enteric Fever Cases and HlyE Seroincidence.** The height of each bar indicates the monthly number of blood-culture-confirmed enteric fever cases. The orange line represents the monthly estimated HlyE seroincidence, with each participant assigned to the month of the midpoint between their baseline and follow-up samples. The top, middle, and bottom panels correspond to the Bangladesh, Malawi, and Nepal STRATAA sites, respectively. Seroincidence estimates based on fewer than 50 participants are suppressed (April and June 2018 in Bangladesh and Malawi, respectively).
(PNG)

**S14 Fig. Seroincidence by Maximum Time Between Samples.** Each panel corresponds to the specific antigen target which was used to classify participants' infection status when calculating seroincidence. Seroincidence estimates for the Bangladesh, Malawi, and Nepal study sites appear in the top, middle, and bottom row of panels, respectively. Solid lines denote the median seroincidence (y-axis) in each age group (x-axis). Vertical lines represent the 95% credible intervals of the seroincidence estimates. Black lines correspond to the primary analysis, which excluded participants with sample pairs collected over 150 days apart, while red lines correspond to a sensitivity analysis which only included participants with < 100 days between samples.
(PNG)

**S15 Fig. Effect of Small Absolute Increases in IgG on Seroincidence Estimates.** Each panel corresponds to the specific antigen target that was used to classify participants' infection status when calculating seroincidence. Seroincidence estimates for the Bangladesh, Malawi, and Nepal study sites appear in the top, middle, and bottom row of panels, respectively. Solid lines denote the median seroincidence (y-axis) in each age group (x-axis). Vertical lines represent the 95% credible intervals of the seroincidence estimates. Black lines correspond to the primary analysis, while red lines correspond to a sensitivity analysis in which participants with a lower IgG at follow-up than a plate- and antigen batch-specific negative control are automatically classified as uninfected, regardless of the relative change in IgG between samples.
(PNG)

**S16 Fig. Seroincidence by Likelihood Function.** Each panel corresponds to the specific antigen target which was used to classify participants' infection status when calculating seroincidence. Seroincidence estimates for the Bangladesh, Malawi, and Nepal study sites appear in the top, middle, and bottom row of panels, respectively. Solid lines denote the median seroincidence (y-axis) in each age group (x-axis). Vertical lines represent the 95% credible intervals of the seroincidence estimates. Black lines correspond to the primary analysis, while red lines correspond to a sensitivity analysis in which the likelihood function for seroincidence directly incorporates participants' mixture model-derived probability of infection.
(PNG)

**S17 Fig. Vi Seroincidence by Age, Study Site, and Assay.** Each panel corresponds to a different study site. Lines denote the estimated seroincidence (y-axis) in each age group (x-axis). Solid lines represent seroincidence estimates based on the serologic data generated in this study with a bead-based multiplex assay, while the dashed lines depict previously published seroincidence estimates based on ELISA assays. Green, orange, and purple lines correspond to the Bangladesh, Malawi, and Nepal study sites, respectively.
(PNG)

## Acknowledgments

We acknowledge the contributions of the participants who took part in the STRATAA study and the large field and laboratory teams at the three study sites who make up the STRATAA Study Group, including Amit Aryja, Binod Lal Bajracharya, David Banda, Yama Mujadidi, Pallavi Gurung, Arifuzzaman Khan, Clemens Masesa, Tikhala Makhaza Jere, Archana Maharjan, George Mangulenji, Maurice Mbewe, Harrison Msuku, Nirod Chandra Saha, Prasanta Kumar Biswas, Anup Adhikari, Md Amiruli Islam Bhuiyan, Christoph Blohmke, Sabina Dongol, Jennifer Hill, Nhu Tran Hoang, Rose Nkhata, Sadia Isfat Ara Rahman, Nazia Rahman, Neil J. Saad, Richard Wachepa, and the Nepal Family Development Foundation team.

## Author contributions

**Conceptualization:** Jo Walker, Andrew J. Pollard, Stephen Baker, James E. Meiring, Merryn Voysey, Virginia E. Pitzer.
**Data curation:** Jo Walker, Susana Camara, Mila Shakya, Deus Thindwa, James E. Meiring, Merryn Voysey.

**Formal analysis:** Jo Walker.

**Investigation:** Jo Walker, Paula Russell, Leanne Kermack, Tan Trinh Van, Tran Vu Thieu Nga, Elli Mylona, Susana Camara, Young Chan Kim, Sonu Shrestha, Arne Gehlhaar, Josefin Bartholdson Scott, Farhana Khanam, Mila Shakya, Deus Thindwa, Melita A. Gordon, Buddha Basnyat, John D. Clemens, Firdausi Qadri, Robert S. Heyderman, Christiane Dolecek, Susan Tonks, Thomas C. Darton, Andrew J. Pollard, Stephen Baker, James E. Meiring, Merryn Voysey, Virginia E. Pitzer.

**Methodology:** Jo Walker, Paula Russell, Leanne Kermack, Tan Trinh Van, Tran Vu Thieu Nga, Elli Mylona, Josefin Bartholdson Scott, Stephen Baker, James E. Meiring, Merryn Voysey, Virginia E. Pitzer.

**Project administration:** Farhana Khanam, Melita A. Gordon, Buddha Basnyat, John D. Clemens, Firdausi Qadri, Susan Tonks, Andrew J. Pollard, Stephen Baker, Merryn Voysey.

**Supervision:** Susana Camara, Farhana Khanam, Mila Shakya, Melita A. Gordon, Buddha Basnyat, Firdausi Qadri, Christiane Dolecek, Andrew J. Pollard, Stephen Baker, James E. Meiring, Merryn Voysey, Virginia E. Pitzer.

**Visualization:** Jo Walker.

**Writing – original draft:** Jo Walker.

**Writing – review & editing:** Jo Walker, Susana Camara, Young Chan Kim, Farhana Khanam, Melita A. Gordon, John D. Clemens, Firdausi Qadri, Robert S. Heyderman, Susan Tonks, Thomas C. Darton, Andrew J. Pollard, James E. Meiring, Merryn Voysey, Virginia E. Pitzer.

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
