## [Decision Letter · Decision Letter 0]

21 May 2025

PNTD-D-25-00483

Leveraging paired serology to estimate the incidence of typhoidal Salmonella infection in the STRATAA study

Dear Dr. Walker,

Thank you for submitting your manuscript to PLOS Neglected Tropical Diseases. After careful consideration, we feel that it has merit but does not fully meet PLOS Neglected Tropical Diseases's publication criteria as it currently stands. Therefore, we invite you to submit a revised version of the manuscript that addresses the points raised during the review process. Kindly also add line and page numbers to your manuscript.

Please submit your revised manuscript within 60 days Jul 20 2025 11:59PM. If you will need more time than this to complete your revisions, please reply to this message or contact the journal office at plosntds@plos.org. Please include the following items when submitting your revised manuscript:

We look forward to receiving your revised manuscript.

Kind regards,

Benedikt Ley, PhD

Guest Editor

Stuart Blacksell

Section Editor

Shaden Kamhawi

co-Editor-in-Chief

Paul Brindley

co-Editor-in-Chief

**Journal Requirements:**

4) We notice that your supplementary Figures are included in the manuscript file. Please remove them and upload them with the file type 'Supporting Information'. Please ensure that each Supporting Information file has a legend listed in the manuscript after the references list.

5) Please amend your detailed Financial Disclosure statement. This is published with the article. It must therefore be completed in full sentences and contain the exact wording you wish to be published. Please ensure that the funders and grant numbers match between the Financial Disclosure field and the Funding Information tab in your submission form. Note that the funders must be provided in the same order in both places as well.

**Reviewers' Comments:**

Reviewer's Responses to Questions

**Key Review Criteria Required for Acceptance?**

**Methods:**

-Are the objectives of the study clearly articulated with a clear testable hypothesis stated?

-Is the study design appropriate to address the stated objectives?

-Is the population clearly described and appropriate for the hypothesis being tested?

-Is the sample size sufficient to ensure adequate power to address the hypothesis being tested?

-Were correct statistical analysis used to support conclusions?

-Are there concerns about ethical or regulatory requirements being met?

Reviewer #1: The objectives of the study are well articulated and the statistical methods and sample size are appropriate.

Were paired samples from the same individuals run at the same time (eg on the same plate) on the Luminex?

For the sensitivity analysis that evaluated the impact of the time between visits, did the authors also consider an additional sensitivity analysis that removed the restriction on time between visits? Or at least examine the distribution of change in standardized IgG response in the <150d vs >=150days groups? This would be mainly relevant in Malawi which has the largest data loss due to this restriction. Given that you are examining IgG responses, these may persist at elevated levels for long periods of time, especially in highly endemic settings with repeated exposures.

Were any of the serosurvey participants among the blood-culture confirmed cases and/or the healthcare-seeking survey? Did the serosurvey include any data capture on recent history of febrile symptoms?

Reviewer #2: Yes

Reviewer #3: • In the first paragraph of the methods (Study design and enrollment of participants), were there no census updates in Malawi? To clarify, there were baseline censuses and final censuses at each site, and Nepal and Bangladesh had additional census updates in between? Please make this clearer.

• In “Sample collection and laboratory testing,” please give the proportion of people who provided consent (or not) and who could be located (or not). Also, please provide the percentage of children younger than 6 months old who were included. Can you provide the distribution of ages sampled? Was this age group oversampled? Was sampling done in proportion to age distribution of the population? Also, please rephrase “participants who could be located.”

• Please provide justification for choosing to use the mixture model to address the study question. Why use this model over, for example, an exponential decay model?

• There are often issues with convergence in mixture models. Please provide further model diagnostics for convergence past visual inspection.

• Why did the authors use the Metropolis algorithm over Metropolis-Hastings?

• Please cite R in the text (last paragraph before “Ethics”).

• In the first paragraph of the Results, please (re)state the inclusion criteria for the participants

**Results:**

-Does the analysis presented match the analysis plan?

-Are the results clearly and completely presented?

-Are the figures (Tables, Images) of sufficient quality for clarity?

Reviewer #1: The Methods section describes that particular efforts were made to recruit children <6m into the study -- as such it would be helpful to know how many were included from this age group. Did the authors consider that maternal IgG may have influenced the ability to detect meaningful changes in IgG response and/or offered protection during this early period?

For Figure 3, it may be worthwhile to split the 0y group into <6m and >6m if you have enough sample size.

The authors focus on significant positive changes in standardized IgG response to estimate seroincidence. Could you interpret significant negative changes as indication that the individuals were recently exposed prior to the baseline measurement (eg the lower half of the middle panels in Figure 1)?

There is a statement that YncE was the only antigen target that had the same rank ordering of sites for seroincidence and clinical incidence. However there's statistically no significant difference between Malawi and Nepal for both YncE and Nepal serocincidence (Also for the legend for Supplementary Figure 5 please define the error bars). It is interesting to note that by HlyE, Nepal and Malawi are very similar in all age groups except for 0-4 -- this may be worthy of some discussion.

The site name strip text is cut off in Supplementary Figure 8 (figure just needs to be slightly smaller to fit the page)

Reviewer #2: Yes

Reviewer #3: • Please show the distribution of days between samples (perhaps in a graph). Are there any concerns or biases with having visits too close together? Does the fact that follow-up samples were collected slightly earlier in Bangladesh (vs. later for others) have any implications for the results?

• General comments for figures:

o Please spell out all acronyms (IgG, CrI, PY, etc)

o Suggest using the word “panel” instead of subfigure

• Table 2: shouldn’t the adjusted numbers be CrI (not CI)?

• Table 2: Are the ratios shown using the median estimates? Why not have uncertainty intervals for the ratios too?

• Why use 95% CrIs in Table 2, but report IQR in Table 1 and the text for other estimates?

• In Table 3, please provide further details about the table (i.e., I think this is Pearson correlation?). Why are there columns without Nepal?

• Figure 1: please label the scales in the middle column. Also, no need to capitalize X- and Y-axis (and this isn’t done in Figure 2)

• Figure 2: Can the authors put the legend altogether, instead separated by two panels? Can the authors also add the 95% CrIs for the adjusted incidence estimates?

• Figure 3 & 4: please add labels for the row panels.

• Supplementary Figures 3 & 4: (optional) consider adding a line for each site; or perhaps the best fit line could be a hierarchical model with random slopes (by country)

**Conclusions:**

-Are the conclusions supported by the data presented?

-Are the limitations of analysis clearly described?

-Do the authors discuss how these data can be helpful to advance our understanding of the topic under study?

-Is public health relevance addressed?

Reviewer #1: The results are well discussed and put into appropriate public health context. One minor point is that at in discussion of the disease burden in adults there is the statement: 'infections in this group are also less likely to result in symptomatic disease' --

it may be the case that these infections are either asymptomatic or mildly symptomatic but don't lead to care-seeking. The authors could propose a hypothetical study design that would help elucidate the differences between clinically significant disease vs exposure/infection that results in antibody responses (and potentially associated transmission) as this is a persistent question in the field.

Reviewer #2: Yes

Reviewer #3: • In the first paragraph of the Discussion, why do the findings suggest infections every few years? This is not clear from the results. Can the authors provide further comment on the duration of waning immunity from the model? Does the model extend past the follow-up visits to measure this?

• Can the authors provide more discussion about the differences between antigen targets? And perhaps recommend one?

• Can the authors provide further broader public health implications from this study? Do the findings suggest a recommendation to revisit the guidelines for which age groups to vaccinate? Can the findings be generalized to other settings or diseases? Are there any implications for decision making or typhoid control and prevention?

**Editorial and Data Presentation Modifications?**

Reviewer #1: The last sentence of the clinical surveillance section in methods should read: Code and data ARE available ...

Reviewer #2: Major Comments:

1. Some seroresponses shown in Figure 1 appear to originate from very low baseline antibody levels (log-sFI < -3) with large fold rises. Could these be artifacts of noise (i.e., small values divided by small values)? Please discuss whether a lower limit of detection was applied and whether any such responses were excluded or flagged.

2. Consider reporting or analyzing the median FI rather than the mean. Median FI may be less sensitive to right-skewed distributions and outliers.

3. The Malawi serosurvey spans approximately 18 months and includes two peak transmission seasons, whereas the Bangladesh and Nepal serosurveys only capture one. Given assumptions about constant FOI, could the inclusion of two peak seasons within an 18-month period have overestimated the annual seroincidence in Malawi? I recommend conducting a sensitivity analysis restricting the Malawi sample to a 12-month window to assess whether this alters the rank ordering of seroincidence estimates between sites.

4. The adjustment factors applied to clinical incidence in Nepal appear unusually large in older age groups (e.g., age 10–14: from 0.28 to 5.47). Could potential issues in the adjustment methodology explain the discordance between clinical and serologic incidence in these age groups in Nepal? Please consider discussing this possibility in the limitations section.

5. The use of a 0.5 posterior probability threshold to classify participants as infected versus not infected could be better justified. Was this threshold chosen arbitrarily or based on prior data or validation? Did the authors consider alternative thresholds? A brief discussion of the rationale and impact on seroincidence estimates would be helpful.

6. Assay Quality Control

- Please describe whether blank values were subtracted from the fluorescence intensity (FI) measurements.

- Were results normalized to a positive control or reference standard? If so, please describe the method.

- The manuscript mentions that results were collected as duplicate mean FI; please report the within-plate coefficient of variation (CV) across duplicates. If any samples were repeated across plates, also report the across-plate CVs.

- A description of quality control procedures should be included in the main text or supplement.

Minor Comments:

1.Consider replacing the term “reported incidence” with “clinical incidence (crude and adjusted)” throughout. "Reported" seems to imply passive surveillance, whereas these estimates come from active prospective surveillance studies.

2.The higher seroincidence rates for LPSO2 and O9 in Malawi may be indicative of a higher burden of non-typhoidal Salmonella at this site.

3.The final sentence of the manuscript would be strengthened by highlighting a key implication. For example: “…to reduce the burden of disease” rather than simply “to generate data.”

4. Visualization

- Consider combining Figures 3 and 4 into a single figure showing seroincidence by year of age (0–4) and then by age group.

- For Supplemental Figure 1: Could you include higher resolution in the younger age range (0–4 years) as in Figure 3 to better explore potential maternal antibody effects? Alternatively, modeling antibody responses over continuous age using a generalized additive model (GAM) may provide improved resolution to detect maternal antibody decay.

- For Supplemental Figure 1: Using a free y-axis may improve visibility of differences by site across antigen targets.

Reviewer #3: Introduction:

• Suggestion for wording in paragraph 1 of introduction: “As a result, enteric fever mostly occurs in settings with limited [access to] clean water and sanitation…”

• Suggestion for wording in paragraph 2 of introduction: “The absence of [robust and routine] surveillance data [limits evidence-based decision making regarding typhoid control and prevention without complete data]”

• The third paragraph in the introduction needs some sort of transition; as it is, it seems a little abrupt and out of the blue. It would be helpful to start with a transition sentence offering serosurveillance as a solution to the lack of routine surveillance data.

• The last paragraph of the introduction does not make the objective of the study as clear as it should be (as it is in the abstract). In that paragraph, it would be helpful to see: 1) motivation and rationale for the study, 2) (optional) brief overview of methods, and 3) and a reframing of the objective of the study. This is unclear as it currently is.

Additional minor comments:

• Spell out acronyms the first time they are used: ELISA, EDTA, 95% CrI, IQR.

• Diagnostic “test” instead of diagnostic.

• Blood culture “test” or “collection” instead of blood culture.

• Consistency with spacing between text and citations.

• Suggest using the word “affordable” instead of “cheap” diagnostic in the second paragraph of the introduction.

• No need to italicize citations of authors (i.e., Aiemjoy et al.). Also, make sure that there is a period at the end of “et al.” in every instance.

• Spell out single-digit numbers (optional).

• Please check whether “Moss.Inc” is the correct spelling and punctuation.

• In the “Ethics” section, no need to capitalize “Individual.”

**Summary and General Comments:**

Reviewer #1: The authors present a well designed study that uses several markers of enteric fever infection to estimate seroincidence from paired samples and compares to clinical surveillance data in the same populations. The manuscript is clearly written and the analyses presented are extensive. With minor revisions this manuscript would be ready for publication.

Reviewer #2: This is an excellent and timely manuscript that leverages paired serology to estimate the incidence of typhoidal Salmonella infection in Bangladesh, Malawi, and Nepal. The authors employ a novel application of a multiplex bead assay and mixture model framework to define recent serologic infection, and the comparison of seroincidence with adjusted clinical incidence provides valuable insight into the underlying burden of typhoidal Salmonella infection.

Reviewer #3: In “Leveraging paired serology to estimate the incidence of typhoidal Salmonella infection in the STRATAA study,” the authors use paired blood samples to estimate the seroincidence of typhoid incidence across multiple countries, age groups, and antigen targets. The authors then compare these estimates to adjusted and unadjusted blood-culture-confirmed incidence estimates. This method and these findings fill a much-needed gap in typhoid burden estimates, linking seroincidence to symptomatic healthcare-seeking cases. The study is well-written and the method and findings are robust, and I have mostly minor comments and suggestions and some requests for further justification.

PLOS authors have the option to publish the peer review history of their article (what does this mean? ). If published, this will include your full peer review and any attached files.

**Do you want your identity to be public for this peer review?** For information about this choice, including consent withdrawal, please see our Privacy Policy .

Reviewer #1: No

Reviewer #2: No

Reviewer #3: No

**Figure resubmission:**

**Reproducibility:**



---

## [Decision Letter · Decision Letter 1]

29 Sep 2025

Dear Mx Walker,

We are pleased to inform you that your manuscript 'Leveraging paired serology to estimate the incidence of typhoidal Salmonella infection in the STRATAA study' has been provisionally accepted for publication in PLOS Neglected Tropical Diseases.

Best regards,

Benedikt Ley, PhD

Guest Editor

Stuart Blacksell

Section Editor

Shaden Kamhawi

co-Editor-in-Chief

Paul Brindley

co-Editor-in-Chief

Reviewer's Responses to Questions

**Key Review Criteria Required for Acceptance?**

**Methods**

-Are the objectives of the study clearly articulated with a clear testable hypothesis stated?

-Is the study design appropriate to address the stated objectives?

-Is the population clearly described and appropriate for the hypothesis being tested?

-Is the sample size sufficient to ensure adequate power to address the hypothesis being tested?

-Were correct statistical analysis used to support conclusions?

-Are there concerns about ethical or regulatory requirements being met?

Reviewer #1: The authors have addressed my prior comments and included relevant sensitivity analyses. I have no additional concerns.

Reviewer #3: (No Response)

**Results**

-Does the analysis presented match the analysis plan?

-Are the results clearly and completely presented?

-Are the figures (Tables, Images) of sufficient quality for clarity?

Reviewer #1: The authors have added many supplementary figures that enhance the presentation and depth of their results section.

Reviewer #3: (No Response)

**Conclusions**

-Are the conclusions supported by the data presented?

-Are the limitations of analysis clearly described?

-Do the authors discuss how these data can be helpful to advance our understanding of the topic under study?

-Is public health relevance addressed?

Reviewer #1: The conclusions are supported by the data presented. The authors have also thoroughly detailed limitations of their study.

Reviewer #3: (No Response)

**Editorial and Data Presentation Modifications?**

Reviewer #1: (No Response)

Reviewer #3: (No Response)

**Summary and General Comments**

Reviewer #1: The authors have provided additional clarification and conducted the requested additional sensitivity analyses. I have no further comments and believe the manuscript is ready for publication.

Reviewer #3: The authors have adequately addressed my comments. The manuscript has robust methods and relevant findings, and reads very well.

PLOS authors have the option to publish the peer review history of their article (what does this mean? ). If published, this will include your full peer review and any attached files.

**Do you want your identity to be public for this peer review?** For information about this choice, including consent withdrawal, please see our Privacy Policy .

Reviewer #1: No

Reviewer #3: No

---

## [Editor Report · Acceptance letter]

Dear Mx Walker,

We are delighted to inform you that your manuscript, "Leveraging paired serology to estimate the incidence of typhoidal Salmonella infection in the STRATAA study," has been formally accepted for publication in PLOS Neglected Tropical Diseases.

Best regards,

Shaden Kamhawi

co-Editor-in-Chief

Paul Brindley

co-Editor-in-Chief
